# Detection of Intermittent Claudication from Smartphone Inertial Data in Community Walks Using Machine Learning Classifiers

**DOI:** 10.3390/s23031581

**Published:** 2023-02-01

**Authors:** Bruno Pinto, Miguel Velhote Correia, Hugo Paredes, Ivone Silva

**Affiliations:** 1INESC Technology and Science, 4200-465 Porto, Portugal; 2Faculdade de Ciências, Universidade do Porto, 4169-007 Porto, Portugal; 3Faculdade de Engenharia, Universidade do Porto, 4200-465 Porto, Portugal; 4School of Science and Technology, Universidade de Trás-os-Montes e Alto Douro, 5000-801 Vila Real, Portugal; 5Angiology and Vascular Surgery, Centro Hospitalar Universitário do Porto, 4099-001 Porto, Portugal; 6Instituto de Ciências Biomédicas Abel Salazar, Universidade do Porto, 4050-313 Porto, Portugal

**Keywords:** peripheral arterial disease, intermittent claudication, inertial data, gait analysis

## Abstract

Peripheral arterial disease (PAD) causes blockage of the arteries, altering the blood flow to the lower limbs. This blockage can cause the individual with PAD to feel severe pain in the lower limbs. The main contribution of this research is the discovery of a solution that allows the automatic detection of the onset of claudication based on data analysis from patients’ smartphones. For the data-collection procedure, 40 patients were asked to walk with a smartphone on a thirty-meter path, back and forth, for six minutes. Each patient conducted the test twice on two different days. Several machine learning models were compared to detect the onset of claudication on two different datasets. The results suggest that we can identify the onset of claudication using inertial sensors with a best case accuracy of 92.25% for the Extreme Gradient Boosting model.

## 1. Introduction

Peripheral arterial disease (PAD) is an occlusive atherosclerotic disease that affects blood vessels and reduces blood flow in the lower limbs [1]. In 2010, Fowkes et al. [2] estimated that around 200 million people worldwide suffered from it, with a significant number of older people affected.

The most common symptom of PAD is intermittent claudication (IC), a sensation of aching, burning, heaviness, or tightness in the muscles of the legs that usually begins after walking a certain distance, walking up a hill, or climbing stairs. It goes away after resting for a few minutes, and this discomfort contributes to a sedentary lifestyle, decreasing physical activity, and aggravating cardiovascular risk factors. In extreme cases, PAD can lead to amputations resulting from ulcerations (tissue damage) or gangrene (tissue death) [3]. Besides the continuous deterioration of physical capacity, such as slower walking speed and shorter distances, social relationships are also impacted because people lose their independence and feel bad asking for help more often from family and friends.

There are various possible treatments to help reduce the effects of PAD, ranging from pharmaceuticals to surgery. However, the most recommended treatment is Supervised Exercise Therapy (SET) [4]. Even though surgery is a faster solution to diminish the IC symptoms, Hiatt et al. [5] preliminary results suggested that SET produces a better long-term improvement in mean claudication and maximum walking distances in patients suitable for surgery than the surgery itself.

Commonly, randomized trials have used treadmill walking performance as the primary outcome measure in clinical trials of therapeutic interventions in patients with PAD. However, there are significant limitations to treadmill walking as an outcome because walking on a treadmill is not representative of walking in daily life. Furthermore, there is a considerable learning effect associated with treadmill walking in which even patients with PAD who did not receive therapeutic interventions experience improved treadmill walking over time. Furthermore, in a study by McDermott et al. [6] involving 156 persons with PAD who completed a six-minute walk test and a treadmill test, the six-minute walk test correlated more closely with physical activity levels in the community than treadmill testing.

Thus, promoting exercise training in patients with PAD and IC is a crucial non-pharmacological and non-surgical strategy to treat this disease, which provides favorable systemic vascular effects that may reduce cardiovascular events and improve blood perfusion. Even better, supervised exercise programs have higher efficacy in improving the physical fitness and the symptomatology of this disease compared to unsupervised exercise programs [7,8,9,10].

An outpatient exercise program allows patients to go on community walks advised by health specialists while being monitored via information and communication technologies. This program is a cost-effective alternative for the supervision of physical exercise for people with PAD. However, evaluating the patient’s improvement only through dialogue and questionnaires can be difficult. For that purpose, a way to assess gait metrics is required. The main contribution of this research is the development of a system for the automatic detection of the onset of claudication based on the analysis of data from the patient’s smartphones. Gait event detection and feature extraction are pre-processing stages to build, respectively, the dataset and the features space. Several machine learning models were designed and evaluated to automatically detect the moment when IC occurs while walking from features calculated from 10-second overlapping windows of the inertial data from a smartphone.

The remainder of the paper is structured as follows. Section 2 explains the WalkingPAD project’s experimental approach for dataset gathering, as well as the computational methods proposed to extract the information needed for IC identification. Section 3 presents the results of the pre-processing stages as well as of several machine learning methods. These results are discussed in Section 4. Section 5 concludes and suggests potential directions for future work.

## 2. Materials and Methods

### 2.1. Participants

The WalkingPAD project aims to collect data from 120 patients with PAD monitored clinically at the Centro Hospitalar e Universitário do Porto, Hospital de Santo António, Portugal. For this exploratory study, we collected data from 40 patients (age 64.18 ± 7.15 years, body mass 78.76 ± 10.96 kg). The patients’ PAD condition was also assessed and characterized using a low to moderate Ankle Brachial Index (ABI) [11] as presented in Table 1. This study followed the ethical code for research with human beings as stated by the Declaration of Helsinki. All participants provided their written informed consent.

### 2.2. Experimental Setup

For the data collection procedure, the patients performed the six-minute walk test (6MWT) [12] by walking at a self-selected speed with a smartphone in the pocket of their trousers on a 30-m path, forward and back. Each patient performed the 6MWT twice on two different days. Furthermore, we asked the patient to indicate when the pain started and to use that information to label the data, so that it would be possible to compare walking patterns before and after the pain sensation.

The smartphone used in this work was a Samsung Galaxy A41 with the inertial measurement unit (IMU) STMicroelectronics LSM6DSL, with a full-scale acceleration range of ±8 g (1 g = 9.8 m/s²) and an angular rate range of ±2000 °/s and resolutions of 0.244 mg and 0.07°/s, respectively, and a YAS549 magnetometer with a range of 2000 μT and resolution of 0.01 μT. The smartphone was placed in the patient’s trousers pocket and data were collected in three planes of motion on the inertial frame: medio-lateral (*X*-axis), vertical (*Y*-axis), and anterior–posterior (*Z*-axis), as shown in Figure 1.

To collect the data, we used the HyperIMU (https://play.google.com/store/apps/details?id=com.ianovir.hyper_imu, accessed on 30 September 2022) application, for the Android system, with a sampling rate of 50 Hz. The data acquired by the application were saved, in a comma-separated-values (CSV) file, for later offline processing. Each row contained a timestamp and three values for each sensor were organized in columns for each axis. The accelerometer data are presented in m/s², the gyroscope data are in rad/s, and the magnetometer data are in μT.

### 2.3. Pre-Processing

#### 2.3.1. Interpolation and Filtering

When the mobile inertial sensors work in power-saving mode, the sampling rate is unstable and depends entirely on the phone’s operating system (OS). The time interval between the two consecutive returned samples is not constant. The sensors generate values only when the forces acting on each dimension have a significant change. Therefore, linear interpolation was applied to the sampling points to ensure equidistant points in time. The resulting data consist of 50 equidistant samples per second.

Inertial sensors’ data samples acquired during the user’s walk inevitably present noise and interference. This additional noise originates from various sources (e.g., idle orientation shifts, screen taps, and bumps on the path). Furthermore, the smartphone’s inertial sensors produce more noise than standalone sensors since its functionalities are entirely governed by the mobile OS. As a result, a digital filter must be applied to limit the influence of noise and interference problems [13].

Fridolfsson et al. [14] investigated the impact of frequency filtering on accelerometer data. They concluded that a low-pass filter with a cutoff frequency of 10 Hz would include all relevant acceleration data for physical activity with minimal effect of noise, since the maximum frequency in steps of a healthy human is about 4 Hz. Therefore, the inertial data were filtered using a fourth-order Butterworth low-pass digital filter with a cutoff frequency of 5 Hz, such that all the relevant data were found below the cutoff point.

#### 2.3.2. Rotation

In previous studies involving activity detection from inertial data, researchers mounted the accelerometers and other sensors in fixed body locations and orientations. We decided to have the patients place their devices in their trouser pockets, as this seemed like the natural location for carrying a mobile phone. We also relaxed the phone’s orientation (screen forward or backward and oriented in portrait or landscape mode) to better simulate actual usage of a mobile phone once the patient would be unlikely to remember to place the phone in the same orientation every time they used it. Therefore, we need to estimate the actual movement on an Earth-fixed reference frame.

The rotation that occurs in the mobile phone inertial frame concerning an external frame can be represented by a combination of the roll, pitch, and yaw rotations (shown in Figure 1b). The roll is the object’s rotation around the *Z*-axis; the pitch is the object’s rotation around its *X*-axis; and the yaw (or heading) of its rotation around its *Y*-axis.

Additionally, when the accelerometer, gyroscope, and magnetometer values are used separately, drift errors appear, particularly in the gyroscope data, lowering the precision in orientation estimation. It is feasible to limit these errors and increase the accuracy, resilience, and consistency of the data acquired by using fusion algorithms that integrate the results of the three sensors. We used an algorithm based on the Madgwick fusion method [15], whereby using the values of each axis of the accelerometer, gyroscope, and magnetometer, we obtain the phone’s orientation in the Earth-fixed frame.

#### 2.3.3. Walking Detection

It was also necessary to delete the data samples in which patients were not walking. Two possibilities were considered for doing so: a threshold model and a machine learning (ML) model. The threshold model is a straightforward solution. A variable or combination of variables and the value that defines the problem are chosen. The ML model is more complex, but it may be solved using simple models such as the random forest or clustering classification [16].

After analysis, we found out empirically that the standard deviation of the acceleration magnitude was less than one m/s² on every sample where the patient was at rest, so we decided on a simple threshold method.

#### 2.3.4. Gait Cycle Event Detection

A gait cycle is defined as two successive steps beginning with one foot touching the ground and ending with the same foot touching the ground again. A search for local minima or maxima is required to detect a cycle. Most studies use positive peaks (local maxima) [17,18,19,20,21,22] to detect the cycles, but negative peaks (local minima) [13,23] can also be used for this purpose. According to Yodpijit et al. [19], the axis parallel to gravity (vertical axis) gives the most precise data to extract the cycle time. Previous research uses the vertical axis to detect gait events [13,24,25]. However, the anterior–posterior axis also shows promising results and is used in many research studies [17,18,19,20,21,22,23].

Another interesting work by Jarchi et al. [26] uses Singular Spectrum Analysis (SSA) combined with the Longest Common Subsequence (LCSS). It incorporates information from all accelerometer axes to estimate parameters, including swing, stance, and stride times. Rather than only detecting local features of the raw signals, the periodicity of the signals is also considered. SSA is a technique that can be applied to time-series data to decompose them into several orthogonal components. These components include varying trends and oscillatory and unstructured noise [27]. SSA has been used in many applications of time series analysis, including denoising and prediction. It has been applied to biosignals such as single-channel respiratory signals, and the source signals are effectively separated [28,29].

Based on the work by Jarchi et al. [26], our algorithm was designed to detect heel contacts and toe-offs and segment gait cycles. The SSA algorithm is first applied to the acceleration signals and decomposes them into orthogonal components. Then we remove the trend and reconstruct the axis signals. The *Y*-axis signal is used to find an interval in the time domain to detect heel contacts. An extrema detection method (https://docs.scipy.org/doc/scipy/reference/generated/scipy.signal.argrelextrema.html, accessed on 30 September 2022) was employed to find the local maximum and minimum points of the *Y*-axis dominant oscillation, which is obtained by reconstructing the signal using the first largest eigenvalue.

The objective of applying the extrema detection method to the first dominant oscillation of the vertical signal is to exploit the signal’s periodicity in detecting right heel contact (RHC) and left heel contact (LHC), not relying only on the explicit peaks in the data. A further validation process is performed after finding all these extremes on the dominant oscillation of the *Y*-axis. To estimate the RHC and LHC more accurately, we constructed a short interval using the corresponding local minima of the dominant oscillation. Then the reconstructed signals on the *Y*- and *Z*-axes are multiplied at each specified interval. The point that gives the minimum value is considered the instant of heel contact.

After finding the heel contacts, the signal of the *Y*-axis is used to determine the corresponding RHC and LHC. First, three local minima indices (given by using the dominant oscillation of the *Y*-axis) are selected. Suppose the mean amplitude value of the signal on the *Y*-axis from the first local minimum index to the second one is greater than the mean value from the second to the third one. In that case, the first local minimum is an RHC, the second one an LHC, and the third one is an RHC. Finally, the Right Toe Off (RTO) event corresponds to the first local minimum peak after the LHC. The Left Toe Off (LTO) is a local maximum before the LHC.

### 2.4. Feature Extraction

After obtaining the walking data with the correct rotation and detecting the gait events, the next step is to create the feature dataset. One approach is to divide the raw data into sequential windows to be processed. In the window approach, its size must be chosen depending on whether the recognition is performed in real-time (or “online”) or not. For online applications, the window size must be defined concurrently with data collection, whereas for offline applications, the window size can be defined before data collection.

The most commonly used approach is the non-overlapping windows technique, where the signal is divided into equal windows with no gaps. However, this approach suffers from the drawback that, as the window size is set arbitrarily, it might split the data in an inconvenient place. It does not capture the "whole cycle" of the activity to be recognized. To avoid this disadvantage, the technique can be used with overlapping, usually 50%, which was the method chosen in our solution.

#### 2.4.1. Time-Domain Features

Time-domain features are simple mathematical and statistical metrics allowing the extraction of basic signal information from raw sensor data. In addition, these metrics are often used as pre-processing steps for metrics in other domains to select key signal characteristics or features.

The following features were calculated for acceleration signals from the IMU, after pre-processing, rotation, and gait cycle detection, and used in the classification methodologies:
**Maximum and Minimum** define the full amplitude or dynamic range of the signal.
**Mean and Median** separate the signal data into two halves.
**Standard deviation** is the square root of the variance and represents both the variability of a dataset and a probability distribution. The standard deviation can indicate the stability of a signal.
(1)$$\sigma =\sqrt{\frac{\sum_{i=1}^{N}\left(x_{i}-\bar{x}\right)}{N-1}}$$
**Distribution histogram** gives us the frequency distribution of numerical data by splitting them into equal-sized bins, in our case, ten bins. 
**Root Mean Square Ratio (RMSR)** is the signal’s Root Mean Square (RMS) ratio in each axis.
(2)$$RMS_{x}=\sqrt{\frac{\sum_{i=1}^{N}\left(x_{i}^{2}\right)}{N}}$$
(3)$$RMS_{T}^{2}=RMS_{X}^{2}+RMS_{Y}^{2}+RMS_{Z}^{2}$$
(4)$$RMSR_{x}=RMS_{x}/RMS_{T}$$
**Signal Magnitude Area (SMA)** is the sum of the area enveloped by the magnitude of each of the three-axis accelerometer signals to compute the energy expenditure in daily activities [30].
(5)$$\frac{1}{N}\sum_{n=1}^{N}|a_{x}\left(n\right)|+|a_{y}\left(n\right)|+|a_{z}\left(n\right)|$$
**Kurtosis** measures the spikiness of the signal.
(6)$$N*\frac{\sum_{i}^{N}{\left(x_{i}-\bar{x}\right)}^{4}}{\sum_{i}^{N}{\left(x_{i}-{\bar{x}}^{2}\right)}^{2}}$$
**Skewness** measures the asymmetry of the probability distribution.
(7)$$g=\frac{m_{3}}{m_{2}^{3/2}}$$
where
(8)$$\;m_{i}=\frac{1}{N}\sum_{n=1}^{N}{\left(x\left[n\right]-\bar{x}\right)}^{i}$$
**Similarity** is the similarity between gait cycle time series calculated from the Longest Common Subsequence (LCSS) algorithm as in [26].
**Cadence** is the number of steps per unit time (usually minutes).
**Stride time** is the duration of each gait cycle. The time interval between two successive steps.
**Swing/Stance ratio** is the difference between the percentage of time of stride and stance. This metric helps to identify the gait cycle’s symmetry. From the literature, we know that the typical value for this ratio to be around 60% [31]. Table 2 summarizes the time-domain features calculated from the rotated acceleration magnitude, ||a||, and individual *x*, *y*, and *z* components ($a_{x},a_{y},a_{z}$) captured from the smartphone’s IMU for each 10-s interval of gait data.

#### 2.4.2. Frequency-Domain Features

Frequency-domain techniques capture the repetitive nature of a signal. This repetition often correlates to the periodic nature of a specific activity, such as walking or running. A commonly used signal transformation technique is the Fourier Transform, which enables representing in the frequency domain (or spectrum) essential characteristics of a time-based signal, such as its average (or DC component) and dominant frequency components.

The following features were used:

**Power Spectral Density (PSD)** describes the power present in the signal as a function of frequency per unit frequency. It can be calculated by
(9)$$PSD\left(e^{j\omega }\right)=\frac{1}{2\pi N}|\sum_{k=1}^{N}\left(|\right|a_{k}\left|\right|e^{-j\omega k}{)|}^{2},$$
where ω(0≤ω≤π) is the angular frequency, $\left|\right|a_{k}\left|\right|$ (*k* = 1, 2, …, *N*) is the k-sample of the acceleration magnitude, and *N* is the total number of acceleration samples. The maximum, mean, and minimum PSD were calculated using the Welch’s method.

**Entropy** can be computed using the normalized information entropy of the discrete FFT coefficient magnitudes excluding the DC component [32]. Entropy helps to differentiate between signals that have similar energy values but correspond to different activity patterns.**Spectral energy** can be computed as the squared sum of its spectral coefficients normalized by the length of the sample window.

**Fast Fourier Transform (FFT) coefficients** where each coefficient corresponds to the amplitude of the signal in a particular frequency. In our case, we used 40 coefficients that correspond to the 4 Hz range with a resolution of 0.1 Hz.

**Discrete Cosine Transform (DCT) coefficients** correlates the signal with the cosine basis function only, while the FFT uses the complete complex exponential basis function $e^{ix}=cosx+isinx$, and again 40 coefficients were used in the 4 Hz range with a resolution of 0.1 Hz. Table 3 summarizes the frequency-domain features calculated from the magnitude of the rotated acceleration, ||a||, which was captured from the smartphone’s IMU, for each 10-s interval of gait data.

### 2.5. Model Selection

To see which models would perform better, we used the Python library, pyCaret [33]—an open-source, low-code machine learning library that automates machine learning workflows. This end-to-end machine learning model management tool allows one to compare, tune, and save different models.

The workflow starts by setting up a training environment to define a transformation pipeline and the train–test split and validation strategy. It enables us to quickly investigate some of the most popular machine learning models, i.e., CatBoost Classifier, Extra Trees Classifier (ET), Extreme Gradient Boosting (XGBoost), Gradient Boosting Classifier (GBC), Light Gradient Boosting Machine (LightGBM), and Random Forest Classifier (RF). The machine learning models were designed to solve a two-classes problem where the output predicted variable is binary, such that: 0—walking is normal without intermittent claudication; 1—walking is modified with intermittent claudication due to pain in the lower limbs.

The features space consists of 70 time-domain features, described in Section 2.4.1, and 85 frequency-domain features, described in Section 2.4.2, and it was employed as input to all of the machine learning models studied in this work.

The different models used the K-fold cross-validation, a resampling procedure used to evaluate machine learning models. This procedure has a single parameter, *K* (set to 10 in our work), which refers to the number of groups to split a given dataset. It generally results in less biased results than other methods, such as a simple training–testing split. In addition to the 10-fold cross-validation procedure, the same random seed was used to minimize test differences.


**CatBoost classifier**


CatBoost [34] builds upon the theory of decision trees and gradient boosting. Because gradient boosting fits the decision trees sequentially, the fitted trees will learn from the former trees’ mistakes and reduce the errors. Adding a new function to existing ones is continued until the selected loss function is no longer minimized.

CatBoost does not use similar gradient-boosting models in the decision tree growth technique. CatBoost, on the other hand, generates oblivious trees, which are trees generated by enforcing the requirement that all nodes at the same level test the same predictor with the same condition. The oblivious trees enable a simple fitting scheme and efficiency, where the tree structure seeks an ideal solution while avoiding overfitting.


**Extra trees classifier**


In theory, an ET [35] is very similar to a Random Forest Classifier, differing only in how the decision trees in the forest are built. The Extra Trees Forest’s decision trees are built from the original training sample. Next, a random sample of k features from the feature set is provided to each tree at each test node, and each decision tree must select the best feature to divide the data following specified mathematical criteria (typically the Gini Index).


**Extreme gradient boosting**


XGBoost [36] is an optimized distributed gradient-boosted decision tree (GBDT) machine learning library. It provides parallel tree boosting, in which decision trees are generated sequentially. Weights are assigned to all the independent variables, which are then fed into the decision tree, predicting the results. The weights of the variables that the tree incorrectly predicted are increased, and the new values are fed into the second decision tree. Individual predictors and classifiers are then combined to form a more precise model.


**Gradient boosting classifier**


A GBC [37] generates a prediction model using a collection of weak models, most commonly decision trees. During the learning process, successive trees are created. This algorithm builds the first model to predict the value and computes the loss, which is the difference between the result of the first model and the actual value. Following the first step, a second model is built to minimize the loss. This method continues until a suitable outcome is obtained.

The main idea behind gradient boosting is to find new trees that minimize the loss function iteratively. The loss function measures how large are the errors our model makes.


**Light gradient boosting machine**


LightGBM [38] is a gradient-boosting framework that uses tree-based learning algorithms. It is designed to be distributed and to have a faster training speed and better accuracy.

LightGBM’s big difference from other tree-based learning algorithms is that it grows trees vertically, while alternative tree-based learning algorithms grow trees horizontally. This means that LightGBM grows trees leaf-wise while alternative algorithms grow level-wise. It will choose the leaf with the highest delta loss to grow. When growing the same leaf, the leaf-wise algorithm can reduce losses more than a level-wise algorithm.


**Random forest classifier**


RF [39] creates a set of decision trees from a randomly selected subset of the training data, obtains a prediction from each tree, and selects the best solution through average voting. It also provides a very good indicator of the feature’s importance.

Individual decision trees tend to overfit the training data, but random forests can alleviate this problem by averaging the prediction results from different trees. As a result, random forests outperform single decision trees’ predictive accuracy.

Specific parameters of each ML model, as described in [34,35,36,37,38,39], were automatically optimized and set by the pyCaret library tool and used without changes in the experiments. The collected dataset was split into 80% for training and 20% for testing purposes, following two different strategies, as further explained in Section 3.5.

## 3. Results

### 3.1. Interpotation and Filtering Results

Firstly, linear interpolation was applied to guarantee that the sampling points were equidistant in time. The resulting data consist of 50 equidistant values per second.

A fourth-order Butterworth low-pass filter with a cutoff frequency of 5 Hz was also applied to the data. The pertinent information can be found below the cutoff point, as the maximum frequency limit in human walking kinematics is around 4 Hz [14]. Results after filtering can be seen in the example acceleration signal in Figure 2.

### 3.2. Rotation Results

To test the rotation algorithm based on Madgwick sensor fusion method [15], we used data from two one-minute walks in which we alternated the phone’s position every 30 s to see if the algorithm worked properly. In the first test, the phone was placed in the participant’s trousers pocket, with the screen facing backward, and then the phone was rotated vertically, with the screen facing forward. We expected a yaw of roughly 0° in the first 30 s and 180° in the next 30 s of this test. It should be near 0° for the roll, but there may be always a slight tilt in the pocket.

In the second test, the phone was placed horizontally in the trousers pocket with the screen facing forward and then rotated horizontally. We anticipated a roll of roughly 90° in the first 30 s of this test and −90° in the next 30-s interval. The yaw angle should be around 180°. The results of these tests were as expected, as shown in Figure 3 and Figure 4.

### 3.3. Gait Detection Results

For the gait detection, the algorithm described in Section 2.3.4 was applied, and an example can be found in Figure 5. Vertical lines indicate the gait cycle events RHC, LHC, RTO, LTO, and each gait cycle is between two consecutive RHC events. Figure 5 depicts the real acceleration signals captured during one of the patients’s walks, and therefore, gait events do not all have the exact same amplitudes or intervals. Intra-subject variability as well as noise effects may lead to spurious, short-term disturbances in the detection of gait events, as can be seen in the different intervals between LHC and RTO events detections.

### 3.4. Data Selection Results

The 40 patients performed 2 trials of the 6MWT, but 11 trials were excluded due to acquisition difficulties, yielding a total of 69 successful trials. For each person, the 6MWT were divided into 10-s data windows with 50% overlap, which resulted in a total of 71 data windows per test and a total of 71×69=4899 windows of walking inertial data (at a sampling rate of 50 samples per second). Furthermore, no gait cycles were detected on 823 data windows, reducing the available data to 4076 walking data windows, each of 10-s duration. Each of these 10-s windows of walking inertial data was manually evaluated and labelled by healthcare professional experts, either as 0–“No IC present” or 1–“IC present” according to the patients’ complaints and behaviour. A total of 1590 data windows were labeled as “No-IC present” and 2486 data windows as “IC-present”, resulting in a data balance of 39% versus 61%.

### 3.5. Comparison of Machine Learning Models

The first step was to perform a comparison of the models that pyCaret supports. In a pyCaret session, we indicated the training and testing data, the target variable, and the strategy used for the 10-fold cross-validation. We conducted two experiments, one where the entire dataset of 4076 walking data 10-second windows were split randomly into training (80%) and testing (20%) sets, and another with the data windows grouped by patient, and then each patient’s complete data were assigned randomly to the training set or to the test set. The purpose was to verify if and how the models generalize to new patients not included in the training set.

The most common classifier performance metrics were evaluated for each model, with respect to the experts’ labels assumed as ground truth, namely, Accuracy, Area Under the Receiver Operating Characteristics Curve (AUC), Recall, Precision, and F1-score.

The results from the first experiment can be seen in the following tables: Table 4 shows the results on the training dataset, while Table 5 shows the results on the test dataset with the best scores in bold.

We also performed the second experiment with a different split strategy. This time, to see how the models would perform on a patient they had never seen before, the split between training and test sets was made based on the patients’ identifiers. As can be seen in Table 6, the results on the training dataset were similar to the previous experiment. However, when looking at the test dataset results, Table 7, we can see a reduction in performance of around 20%, which indicates that the models do not generalize well.

## 4. Discussion

The main goal of this study was to develop a method for automatically detecting the onset of claudication using data collected from patients’ smartphones. From the 80 (6-min) walks conducted, we could only use data collected from 69 walks. Each walk was segmented using a 10-s interval with 50% overlap, yielding 71 data windows per walk. After all the data pre-processing, segmentation, and selection, a total of 4076 10-s data windows of the 6MWT were used in feature extraction to calculate 155 features for each data window. Using the python library pyCaret, we compared several machine learning models, namely CatBoost, ET, XGBoost, GBC, LightGBM, and RF, to identify the onset of claudication.

In our first experiment, we used a simple train/test split, and we obtained the best results with the XGBoost, with an overall accuracy of 92.25% on the test dataset, closely followed by LightGBM, which differs only below the hundredths on all metrics and has a slightly higher AUC score. These results suggest that models based on optimized gradient boosting algorithms are more efficient for this problem compared to other tree-based learning models. However, the RF model shows a higher (>3%) recall score, suggesting that RF would be the best model to choose to avoid false negatives, since recall measures the sensitivity of the model. Overall, there is no consistent decrease in the performance metrics between training and testing datasets, which appears to suggest that there is no overfit on the training of any of the models under these experimental conditions.

Another experiment was conducted where the split method was altered to test the generalizability of this model. This time, the split was determined by patient identifier rather than a blind split of the 10-s walk data windows, allowing us to observe how the models would respond to a patient they had never seen before. For the training dataset, we obtained similar accuracy, even with an increase of around 3–5%, with the models based on optimized gradient boosting algorithms, CatBoost, XGBoost, and LightGBM, showing better overall performance metrics with accuracy above 95%. However, in the test dataset, the results obtained were much lower, with a 74.66% accuracy for the best model, and CatBoost Classifier apparently being less prone to overfitting than the others. These results prove that the models do not generalize well for new patients and rely on individual intra-subject characteristics to detect the presence of IC. In other words, if we do not train the models with walking data from one specific patient, it will be harder to detect IC on that patient from what the models learned with the patients in the training dataset.

The performance metrics results demonstrate the applicability of the chosen classification models for predicting the onset of claudication. However, a smaller set of performance metrics, depending on the pursued objectives, must be selected before the classifiers can be deployed.

Nevertheless, the present case study and its findings have some limitations. As was already mentioned, this dataset is limited in size. In contrast to many other application domains of machine learning models, the amount of data in the present context of the WalkingPAD project was relatively small, with only 69 six-minute walks from 40 different patients. The same criteria apply to the quality and quantity of features. Many of the selected features may not have direct relationships with clinical gait characteristics of patients with PAD. Further research and extensive work on feature selection analysis and tuning would be required to obtain explainable and interpretable ML models with respect to clinical gait changes due to the presence of IC.

## 5. Conclusions

We present in this article a pipeline to acquire and process inertial data to obtain temporal and frequency gait cycle features and evaluate the onset of claudication in patients with PAD. The results show that we can detect the onset of claudication based on inertial sensors from a smartphone, with a best case accuracy of 92.25% for the Extreme Gradient Boosting model. However, results also show that the tested ML models do not generalize well for new patients, with a best case accuracy for the CatBoost Classifier of 74.66%. Further research is required to determine which features are most significant for the models and relevant to explain clinical gait characteristics in PAD patients. Furthermore, with the growth of the dataset, different techniques should be investigated, such as the use of deep learning models. In the future, a real-time solution should be implemented to test how the model will act in a less controlled environment.

## Figures and Tables

**Figure 1 sensors-23-01581-f001:**
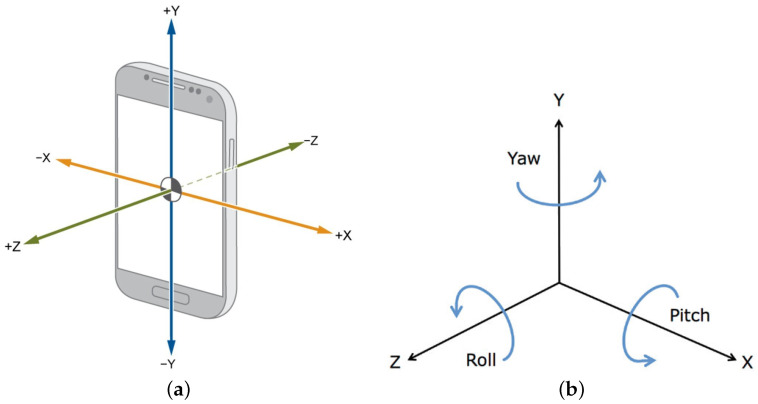
(**a**) Inertial frame orientation of the triaxial IMU in the smartphone; (**b**) Earth-fixed reference frame.

**Figure 2 sensors-23-01581-f002:**
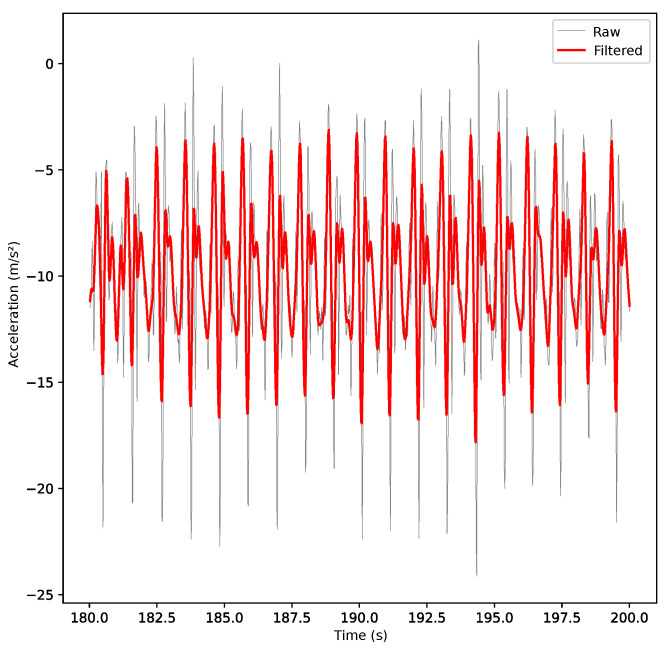
Example of raw data signal (grey) overlaid with filtered data signal (red), after applying a fourth-order low-pass Butterworth filter with cutoff frequency of 5 Hz.

**Figure 3 sensors-23-01581-f003:**
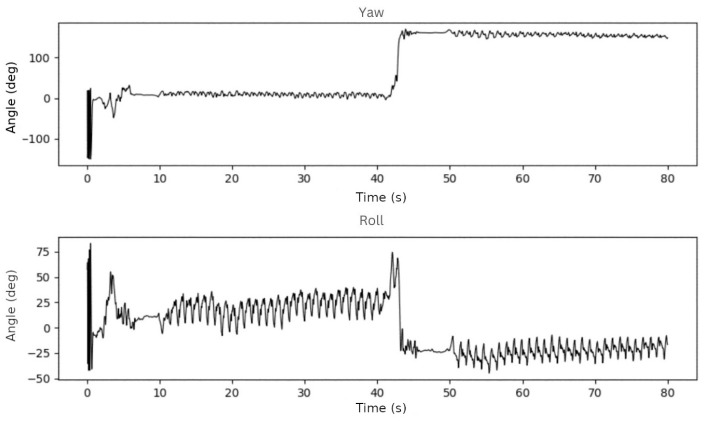
Yaw and roll angles on the first test.

**Figure 4 sensors-23-01581-f004:**
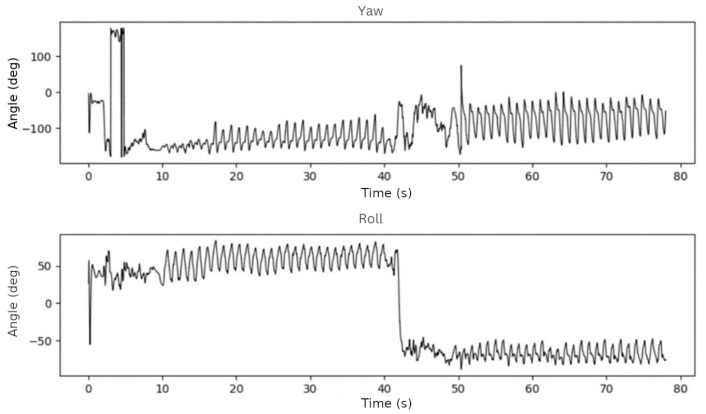
Yaw and roll angles on the second test.

**Figure 5 sensors-23-01581-f005:**
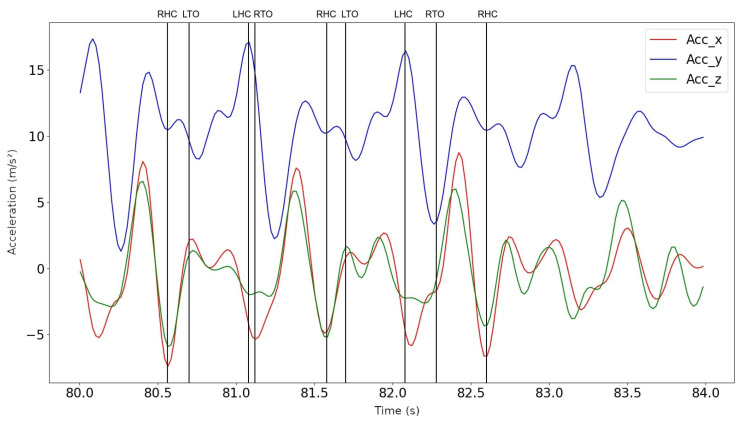
Gait Cycle Detection Example. Each gait cycle is between two consecutive right heel contact (RHC) events.

**Table 1 sensors-23-01581-t001:** Description of patients.

Parameter	Min	Max	Mean	Std
Age (years)	52	78	64.18	7.15
Weight (kg)	57	109.3	78.76	10.96
ABI Right limb Pre Exercise	0.35	1.04	0.71	0.19
ABI Left limb Pre Exercise	0.39	1.07	0.7	0.17
ABI Right limb Post Exercise	0.24	1.02	0.66	0.19
ABI Left limb Post Exercise	0.2	1.07	0.63	0.21

**Table 2 sensors-23-01581-t002:** Time-domain features.

Feature	Signals	Nr. of Coefficients
Maximum	$||a||,a_{x},a_{y},a_{z}$	4
Minimum	$||a||,a_{x},a_{y},a_{z}$	4
Mean	$||a||,a_{x},a_{y},a_{z}$	4
Median	$||a||,a_{x},a_{y},a_{z}$	4
Standard Deviation	$||a||,a_{x},a_{y},a_{z}$	4
Distribution histogram	$||a||,a_{x},a_{y},a_{z}$	40
Root Mean Square Ratio	$a_{x},a_{y},a_{z}$	3
Signal Magnitude Area	$a_{x},a_{y},a_{z}$	1
Kurtosis	||a||	1
Skewness	||a||	1
Similarity	||a||	1
Cadence	$a_{y}$	1
Stride time	$a_{y}$	1
Swing/Stance ratio	$a_{y}$	1

**Table 3 sensors-23-01581-t003:** Frequency-domain features.

Feature	Nr. of Coefficients
Power Spectral Density	3
Entropy	1
Spectral energy	1
FFT coefficients	40
DCT coefficients	40

**Table 4 sensors-23-01581-t004:** Training dataset results (best scores in bold).

Model	Accuracy	AUC	Recall	Precision	F1-Score
CatBoost Classifier	0.9231	0.9788	0.9427	0.9315	0.9370
Extra Trees Classifier	0.8531	0.9416	0.9406	0.8374	0.8858
Extreme Gradient Boosting	0.9222	0.9763	0.9389	0.9334	0.9360
Gradient Boosting Classifier	0.8744	0.9509	0.9168	0.8808	0.8983
Light Gradient Boosting	**0.9254**	**0.9791**	0.9384	**0.9387**	**0.9385**
Random Forest Classifier	0.8803	0.9513	**0.9562**	0.8615	0.9063

**Table 5 sensors-23-01581-t005:** Test dataset results (best scores in bold).

Model	Accuracy	AUC	Recall	Precision	F1-Score
CatBoost Classifier	0.9185	0.9768	0.9433	0.9272	0.9352
Extra Trees Classifier	0.8567	0.9460	0.9449	0.8439	0.8915
Extreme Gradient Boosting	**0.9225**	0.9752	0.9339	**0.9413**	**0.9375**
Gradient Boosting Classifier	0.8508	0.9417	0.8992	0.8665	0.8825
Light Gradient Boosting	0.9215	**0.9774**	0.9323	0.9412	0.9367
Random Forest Classifier	0.8881	0.9590	**0.9638**	0.8706	0.9148

**Table 6 sensors-23-01581-t006:** Split by patient training set results (best scores in bold).

Model	Accuracy	AUC	Recall	Precision	F1-Score
CatBoost Classifier	**0.9521**	**0.9908**	**0.9646**	**0.9566**	**0.9605**
Extra Trees Classifier	0.8805	0.9559	0.9460	0.8687	0.9055
Extreme Gradient Boosting	0.9508	0.9888	0.9624	0.9565	0.9594
Gradient Boosting Classifier	0.9131	0.9702	0.9444	0.9148	0.9293
Light Gradient Boosting	0.9511	0.9893	0.9630	0.9565	0.9597
Random Forest Classifier	0.9054	0.9718	0.9561	0.8948	0.9244

**Table 7 sensors-23-01581-t007:** Split by patient test set results (best scores in bold).

Model	Accuracy	AUC	Recall	Precision	F1-Score
CatBoost Classifier	**0.7466**	**0.8072**	0.7718	**0.8156**	**0.7931**
Extra Trees Classifier	0.6008	0.5694	0.7248	0.6687	0.6957
Extreme Gradient Boosting	0.7181	0.7781	0.7466	0.7932	0.7692
Gradient Boosting Classifier	0.7254	0.7731	0.7651	0.7917	0.7782
Light Gradient Boosting	0.7212	0.7813	0.7282	0.8097	0.7668
Random Forest Classifier	0.7170	0.7487	**0.7936**	0.7654	0.7792

## Data Availability

The data presented in this study are available on request from the corresponding author. The data are not publicly available due to data privacy protection regulations currently in force.

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
