# Peer review of "Detection of Intermittent Claudication from Smartphone Inertial Data in Community Walks Using Machine Learning Classifiers"

_sensors, 2023, doi:10.3390/s23031581_

Round 1

Reviewer 1 Report

Dear authors,

Congratulations on your interesting article comparing six supervised machine learning models trained to identify the occurrence of claudication in people with peripheral arterial disease. According to Tables 2 and 3, which present performance metrics such as accuracy, recall, and precision, it can be inferred that the predicted variable is categorical; however, it is necessary to answer the following questions: (1) how many levels were used for the predicted variable? (2) how were these levels defined? (3) and for the training phase, how the values of the predicted variable were obtained, (4) the levels of the predicted variable were defined according to the five ABI categories (normal, acceptable, some arterial disease, moderate arterial disease, and severe arterial disease) or authors combined two or more categories? If the authors used three or more categories a complete confusion matrix should be included. Regarding the predictors, the paper should specify which of the variables presented in Sections 2.4.1 and 2.4.2 were used as the predictors of the machine learning models.

Author Response

Thank you for your careful review of our manuscript, and for the comments that helped us to improve its quality.

Replying to your specific questions: (1) the predicted variable is actually binary, so only two levels were used for it. (2) the two binary levels are defined below. (3) For the training phase, each 10-seconds window of walking inertial data was manually evaluated and labeled by healthcare professional experts as explained below, (4) the ABI categories were not used nor combined, and since there are only two classes for the predicted variable, we believe there is no need to present the confusion matrices for all the six models, since it would duplicate the information conveyed by the performance metrics. We included the following paragraphs for better clarification of the methods’ performance metrics results and predicted variable:

In section 2.5: The machine learning models were designed to classify in a two classes problem where the output predicted variable is binary, such that:

0 – walk is normal without Intermittent Claudication;

1 – walk is modified with Intermittent Claudication due to pain in the lower limbs.

In section 3.4: The 40 patients performed 2 trials of the 6MWT, but 11 trials were excluded due to acquisition difficulties, yielding a total of 69 successful trials. For each person, the 6MWT were divided in 10-second data windows with 50% overlap, which results in a total of 71 data windows per test and a total of 71 x 69 = 4899 windows of walking inertial data (at a sampling rate of 50 samples per second). Each of these 10-seconds windows of walking inertial data was manually evaluated and labeled by healthcare professional experts, either as 0--"No IC present" or 1--"IC present" according to the patients complaints and behaviour. Furthermore, no gait cycles were detected on 823 data windows, reducing the available data to 4076 walking data windows, each of 10-second duration.

As for the predictors of the machine learning models, the variables presented in sections 2.4.1 and 2.4.2 collectively form the features space. For better clarification of the variables used in the machine learning model, we included the following paragraph:

In section 2.5: “The features space consists of 70 time-domain features, described in section 2.4.1, and 85 frequency-domain features, described in section 2.4.2, and it was employed as input to all of the machine learning models studied in this work.

Other enhancements were made to clarify and improve readability, highlighted in the revised manuscript.

Reviewer 2 Report

This study detected intermittent claudication using smartphone inertial data on overground walking of PAD patients. The author presented his argument well through various machine learning methods. Unfortunately, it seems that the manuscript will not be enough for publication.

1)     A general increase in the level of language would be helpful.  It is recommended that the authors seek out a native English-speaking writer to enhance readability and clarity.

2)     The purpose of this manuscript is not clearly explained. As far as I understand, the purpose of this study is to detect IC of PAD patients using inertial data of smartphone on overground walking. However, there are only descriptions of gait event detection, feature extraction, and classification from the inertial data, but it does not explain how to detect IC. Did you distinguish between patients with IC and those without IC? Or did you detect the moment when IC occurs while walking?

3)     What do you mean by accuracy in 3.5 section (tables 2,3,4,5)? The training dataset is used to train the classifier. What does the accuracy of the training dataset results represent? Also, I suggest explaining by dividing it into inter-subject and intra-subject.

4)     Authors should show the results of applying preprocessing step by step in pictures.

5)     In 2.4 section, I recommend tabulating the time and frequency features.

6)     In 2.5 section, along with a general description of the classifier used, specific parameters used in the actual experiment should be described.

7) Figures 3,4,5 should be clearly presented on x and y-axis.

8)     There is a lack of explanation for clinical data. What is the patient's walking speed? Self-selected or setting speed?

9)     One person walked for 6 minutes and only used 10 seconds of data? With a window length of 10 seconds and an overlap of 50%, did each person collect 950 samples (sampling rate 50/s)? What do you mean of 4076 samples? How organized of the 155 features? A detailed explanation should be required.

Author Response

Thank you for your careful review of our manuscript, and for the comments that helped us to clarify several issues and improve its overall quality.

Replying to your specific questions:

1) the paper is now thoroughly reviewed by a native English-speaking writer.

2) As stated in lines 53-55, you are correct, the purpose is to detect Intermittent Claudication (IC) in Peripheral Arterial Disease (PAD) patients using inertial data of smartphone on overground walking. To clarify this issue, the following paragraph was added:

In section 1, lines 55-59: Gait event detection and feature extraction are pre-processing stages to build, respectively, the dataset and the features space. Several machine learning models were designed and evaluated to automatically detect the moment when IC occurs while walking from features calculated from 10-seconds overlapping windows of the inertial data from the smartphone.

3) The training dataset was manually evaluated and labeled by healthcare professional experts according to the patients complaints and behavior on each walk test. Accuracy in the training dataset exposes the fact that none of the ML methods is perfect and consequently does not achieve 100% performance with respect to the labeled data. Inter-subject and intra-subject evaluations were not considered because each patient performed only two trials of the 6MWT, while a large proportion (80%) of the walking data is required for training the ML models. As stated in lines 353-355, in a second experiment, we were able to split by patient ID and assign different patients to the training dataset and to the test dataset, to evaluate how the models perform on new unseen patients. This can be regarded as an inter-subject evaluation. The following statements were added to the manuscript:

In section 2.5, lines 273-276: The machine learning models were designed to solve a two classes problem where the output predicted variable is binary, such that:

0 – walk is normal without Intermittent Claudication;

1 – walk is modified with Intermittent Claudication due to pain in the lower limbs.

In section 3.4, lines 373-376: Each of these 10-seconds windows of walking inertial data was manually evaluated and labeled by healthcare professional experts, either as 0--"No IC present" or 1--"IC present" according to the patients complaints and behaviour.

In section 3.5, lines 387-389: The most common classifier performance metrics were evaluated for each model, with respect to the experts labels assumed as ground-truth, namely: Accuracy, Area Under the Receiver Operating Characteristics Curve (AUC), Recall, Precision and F1-score.

In section 3.5, lines 393-396: We also performed the second experiment with the different split strategy. This time, to see how the models would perform on a patient they had never seen before, the split between training and test sets was made based on the patients' identifiers.

4) Figures 3, 4, 5 and 6 show signals before and after pre-processing stages. The axes, legends, captions and descriptions were improved and clarified.

5) Two tables were included in section 2.5, table 2 and table 3 summarizing, respectively, the time-domain and the frequency-domain features.

6) Regarding specific parameters for each of the classifiers, described in the references cited in the manuscript, we included the following paragraph:

In section 2.5, lines 339-342: Specific parameters of each ML model, as described in [34-39], were automatically optimized and set by the pyCaret library tool and used without change in the experiments. The collected dataset was split into 80% for training and 20% for testing purposes, following two different strategies as further explained in section 3.5.

7) Axes and axes scales are now clear in Figures 3, 4, and 5.

8) Patients walked at self selected speed. In section 2.2 the following statement was rewritten as:

For the data collection procedure, the patients performed the six-minutes walk test (6MWT) [12] by walking at a self-selected speed with a smartphone in their trousers pocket, on a 30 meters path, forward and back. Each patient performed the 6MWT twice on two different days. Also, we asked the patient to indicate when the pain started, and use that information to label the data, so it is possible to compare walking patterns before and after the pain sensation.”

The following reference was also included regarding the 6MWT:

[12] Ubuane, P.O.; Animasahun, B.A.; Ajiboye, O.A.; Kayode-Awe, M.O.; Ajayi, O.A.; Njokanma, F.O. The historical evolution of the six-minute walk test as a measure of functional exercise capacity: a narrative review. Journal of Xiangya Medicine 2018, 3.

9) Indeed, the collected dataset was not sufficiently well described, thank you for pointing that out. The following clarifying paragraph was added to the manuscript:

In section 3.4, lines 369-377: The 40 patients performed 2 trials of the 6MWT, but 11 trials were excluded due to acquisition difficulties, yielding a total of 69 successful trials. For each person, the 6MWT were divided in 10-second data windows with 50% overlap, which results in a total of 71 data windows per test and a total of 71 x 69 = 4899 windows of walking inertial data (at a sampling rate of 50 samples per second). Each of these 10-seconds windows of walking inertial data was manually evaluated and labeled by healthcare professional experts, either as 0--"No IC present" or 1--"IC present" according to the patients complaints and behaviour. Furthermore, no gait cycles were detected on 823 data windows, reducing the available data to 4076 walking data windows, each of 10-second duration.

Other enhancements were made to clarify and improve readability highlighted in the revised manuscript.

Round 2

Reviewer 1 Report

Since all observations were succesfully addressed, I recommend the paper for publication.

Congratulations.

Author Response

We are deeply grateful for the comments and suggestions that have allowed us to improve the quality of our article.

Thank you.

Reviewer 2 Report

1)     Abbreviations and original words are mixed in the text. And Paragraphs are too many divided.

2)     Axes are still unclear. For example, on the x-axis in figure 4, Timestamps meaning Timestamp/s ? The y-axis is also misleading.

3)     It is better to combine Figure 1 and Figure 2 into one.

4)     It seems that the spacing between the LHC RTO in Figure 6 is too close. Are the experts properly classified or checked?

5)     When evaluating classification accuracy through machine learning, we need to consider the balance of the number of data size. Data balance is very important. What is the ratio of no IC present and IC present data?

6)     It looks like the paper is more about classification than IC detection. It is recommended to edit the title appropriately.

7)     The conclusion of this paper is that IC can be classified through various classifier? Or Does the XG-Boosting classifier perform best in classifying IC? XG Boosting showed the best accuracy in intra-subject test, but not in inter-subject text. Therefore, the conclusion that XG-Boosting is the best is also misleading.

8)     Classification results should be analyzed by discussing how clinical gait characteristics differ with the presence or absence of IC.

Author Response

Thank you again for taking the time to review our article and for the very pertinent and helpful comments that allowed us to further improve it.

Replying to the each specific issue you pointed out:

1)    Abbreviations are now consistently used throughout the text. Several paragraphs continuing  the same topics were joined together to decrease paragraph fragmentation.
2)    Horizontal and vertical axes are now renamed consistently in all figures. 
3)    Figures 1 and 2 are now in a single side by side figure.
4)    Thank you for pointing that out. To clarify the different spacing between the LHC RTO events, the following sentence was added to section 3.3, lines 358-362:“Figure 5 depicts the real acceleration signals captured during one of the subject’s walks and therefore gait events do not all have the exact same amplitudes or intervals. Intra-subject variability as well as noise effects may lead to spurious, short term disturbances in the detection of gait events as can be seen in the different intervals between LHC and RTO events detections.” 
5)    Data balance is now included in the description of the dataset. The following sentence was added in section 3.5, lines 374-375: “A total of 1590 data windows was labeled as “No-IC present” and 2486 data windows as “IC-present”, resulting in a data balance of 39\% versus 61\%.”
6)    The final aim of the work presented is actually to detect the onset of Intermittent Claudication in real-time as the patient walks. However, to highlight the use of ML classification and following the reviewer recommendation, the title was changed to: “Detection of Intermittent Claudication by Classifying Smartphone Inertial Data in Community Walks”
7)    Indeed, for the inter-subject experiment, the best method was the CatBoost Classifier as explained in the Discussion, lines 417-420. Nevertheless, to emphasize this point, the following statement was added to the Conclusions, lines 439-440: “However, results also show that the tested ML models do not generalize well for new patients with a best case accuracy for the CatBoost Classifier of 74.66%.”
8)    This is actually a limitation of this work an a matter for further investigation in future work. The following statement was added to the Discussion section 4, lines 429-433: “Many of the selected features may not have direct relationships with clinical gait characteristics of patients with PAD. Further and extensive work on feature selection analysis and tuning would be required to obtain explainable and interpretable ML models with respect to clinical gait changes due to the presence of IC.”
    and to the Conclusions section 5, lines 441-442: “Further research is required to determine which features are most significant for the models and relevant to explain clinical gait characteristics in PAD patients.”

Round 3

Reviewer 2 Report

1)     Looking at the contents of the manuscript, the following title seems more appropriate. “Classification of Intermittent Claudication using Smartphone inertial data during walking based on machine learning algorithm”.

2)     The discussion section should include an analysis of the results presented, but only the results are listed. For example, the author wrote the sentence as follows in discussion section: “In our first experiment, we used a simple train/test split, and we obtained the best results with the XGBoost, with an overall accuracy of 92.25% on the test dataset. As for the other metrics, they were also better in the XGBoost, except AUC which was better in LightGBM and recall which was higher with the RF model (> 3%), indicating that RF would be the best model to choose to avoid false negatives.” This sentence is only a summary of the results. This sentence would be more appropriate for the results section. The discussion section must contain an analysis of the results. for example, why XGBoost has the highest accuracy in this study? or why is RF the best model to choose to avoid false negatives?

3)     Other sentences in the discussion section should also be revised.

Author Response

Thank you one again for taking the time to further review our manuscript. All your comments were greatly appreciated and taken ointo consideration to improve the article.

Replying to the specific issues:

1) Although we value the reviewer’s opinion, we respectfully disagree on part of the suggested title. In particular, the terms “Classification of Intermitent Claudication” may mislead the reader to believe we are classifying IC, for example, in different levels of severity, which is not the case as we clarified on the two previous reviews. We are simply detecting the presence or absence of IC from unconstrained walking data features with machine learning classifiers. Therefore we propose the following title:

“Detection of Intermitent Claudication from Smartphone Inertial Data in Community Walks Using Machine Learning Classifiers”

2) The purpose of the article is to prove the applicability of machine learning models to classify walking changes due to IC. A deeper discussion on the differences between the ML models, and why one particular model is better than another, is out of scope of the present work and also, we believe, out of scope of the special issue. Nevertheless, to provide a more detailed discussion on the results, we rewrote the sentences as follows:

“In our first experiment, we used a simple train/test split, and we obtained the best results with the XGBoost, with an overall accuracy of 92.25% on the test dataset, closely followed by LightGBM, which differs only below the hundredths on all metrics and with a slightly higher AUC score. These results suggest that models based on optimized gradient boosting algorithms are more efficient for this problem compared to other tree-based learning models. Although, the RF model shows a higher (> 3%) recall score, suggesting that RF would be the best model to choose to avoid false negatives, since recall measures the sensitivity of the model. Overall, there is no consistent decrease on the performance metrics between training and test datasets, which appears to suggest there is no overfit on the training of any of the models under these experimental conditions.”

3) We also revised and rewrote other sentences in the discussion as follows: 

“For the training dataset, we obtained similar accuracy, even with an increase of around 3-5%, with the models based on optimized gradient boosting algorithms, CatBoost, XGBoost and LightGBM, showing better overall performance metrics with accuracy above 95%. However, in the test dataset, the results obtained were much lower, with a 74.66% accuracy for the best model, CatBoost Classifier apparently being less prone to overfitting than the others. These results prove that the models do not generalize well for new patients, and are relying on individual intra-subject characteristics to detect the presence of IC. In other words, if we do not train the models with walking data from one specific patient, it will be harder to detect IC on that patient from what the models learned with the patients in the training dataset.”